# Exploring Vision-Language Alignment under Subtle Contradictions

**Author**

## Abstract

Vision-language models (VLMs) have made notable progress in tasks such as object detection, scene interpretation, and cross-modal reasoning. However, they continue to face significant challenges when subjected to adversarial attacks. The simplicity of including hidden text in websites points to a critical need for a deeper understanding of how misleading text disrupts performance in multimodal applications. In this study, we systematically introduce faintly embedded and clearly visible contradictory text into a large-scale dataset, examining its effects on object counting, object detection, and scene description under varying text visibility. Our findings show that counting accuracy suffers significantly in the presence of adversarial textual perturbations, while object detection remains robust and scene descriptions exhibit only minor shifts under faint disruptions. These observations highlight the importance of building more resilient multimodal architectures that prioritize reliable visual signals and effectively handle subtle textual contradictions, ultimately enhancing trustworthiness in complex, real-world vision-language scenarios.

## 1 Introduction

Large Language Models (LLMs) have driven remarkable progress in diverse textual transformation and generation tasks, offering a powerful foundation for emerging multimodal systems (Jiang et al., 2024; Yonekura et al., 2024). Their integration with computer vision architectures has produced vision-language paradigms for applications like image captioning and scene interpretation (Bitton et al., 2023; Liu et al., 2023). Yet, recent work reveals persistent limitations in managing conflicting inputs across modalities, highlighting a need for more robust solutions (Zhao et al., 2023).

Within the realm of vision-language modeling, contradictory textual prompts have become a key concern (Qraitem et al., 2025; Wang et al., 2024; Cheng et al., 2024). An open question focuses on how faintly embedded versus clearly visible contradictory text disrupts the alignment of visual and textual signals. Many vision-language models exhibit performance declines under conflicting cues but lack thorough investigation into subtle contradictions (Qraitem et al., 2025). Addressing these disruptions is essential for applications requiring accurate object recognition, scene understanding, and robust cross-modal integration (Cheng et al., 2024).

This paper systematically explores the influence of both subtle and overt contradictory text by manipulating text visibility in multiple tasks. We address a gap in current benchmarks by isolating the textual component's role in degrading object counting, visual detection, and descriptive accuracy. Novel methodological choices include precise control of text opacity, ensuring that even faint contradictions can alter vision-language representations. These measures illuminate the degrees of visual-linguistic conflict and inform potential avenues for more robust multimodal architectures. Empirical results indicate that contradictory text markedly decreases counting accuracy, dropping by up to 0.078 as text visibility intensifies. Other tasks, such as cat detection, remain comparatively

stable, underscoring the significance of task-specific cues. By comprehensively evaluating how varying text visibility affects system output, this work reveals key vulnerabilities in vision-language alignment. Its contributions include highlighting the need for better handling of misleading lexicon and introducing frameworks that can guide more resilient future VLM designs.

## 2 Related Works

**Vision-Language Models.** Vision-language models (VLMs) have attracted considerable attention for their capacity to embed and align textual and visual features, enabling tasks such as image captioning, visual question answering, and object detection (Yonekura et al., 2024; Li et al., 2023). Notable architectures integrate large-scale pre-training to learn joint representations that generalize across multiple modalities (Segal et al., 2022; Yang et al., 2024; Bai et al., 2023; Wang et al., 2023a). Despite rapid advances, these works reveal persistent weaknesses when textual inputs conflict with visual cues, underscoring the need for strategies to handle inconsistent information (Cheng et al., 2024; Qraitem et al., 2025).

**Evaluation Metrics and Gaps.** Recent efforts propose expanded benchmarks assessing VLMs under varied instructions and adversarial perturbations (Bitton et al., 2023; Wang et al., 2023b; Bai et al., 2023; Dai et al., 2023; Shirnin et al., 2024). However, few approaches systematically manipulate text visibility to uncover the range of model vulnerabilities. Building on these gaps, this paper examines how faint and visible conflicting text affect inference across multiple tasks, contributing a more nuanced evaluation of model robustness in adversarial settings.

## 3 Methods

We aimed to determine how faintly embedded or clearly visible contradictory textual cues affect a large-scale vision-language model performing visually grounded tasks. Our main hypothesis posited that even subtle contradictions might disrupt object detection, counting, or descriptive accuracy, while more conspicuous text would heighten such disruptions. We were guided by questions around whether the model could discriminate misleading textual information from actual visual cues and how varying degrees of text visibility might alter predictions in tasks such as object enumeration (dogs), object presence (cats), and scene description.

We employed the COCO 2017 training set (Lin et al., 2015), sampling 5000 images to support three tasks: (a) object counting (focusing on dogs), (b) visual search (detecting cat presence), and (c) scene description (identifying objects and colors). Each image was duplicated into three conditions: Original (no text), Faint Text (alpha-blended, near-invisible contradictory text), and Visible Text (clear white font with a black outline). Thus, we aggregated a total of 15,000 image-based data points. We leveraged the `Qwen2.5-VL-7B-Instruct` model (Team, 2024), which processes both images and textual prompts without additional fine-tuning on an NVIDIA A100 GPU. Prompts were customized per task—requesting a count, a yes/no determination, or a compositional scene description.

We gathered performance measures for each task under each condition. For counting, we measured accuracy (perfect dog counts) and mean absolute error (MAE). For visual search, we evaluated accuracy based on correct yes/no recognition of cat presence. The scene description task involved four metrics: object recall, color accuracy, spurious objects, and number of objects mentioned. These metrics offered complementary lenses to understand how contradictory text affects numeric, Boolean, and descriptive outputs.

Alpha-blending in faint text scenarios was carefully tuned so that misinformation was barely visible yet present in the pixel space. In visible text conditions, bold, high-contrast statements were placed in regions of minimal overlap with salient objects. Each modified image was run through the model with standardized prompts, and output parsing was automated to extract dog counts, cat presence, or descriptive text. By systematically varying text visibility while preserving core visual content, we isolated the direct impact of contradictory text on vision-language alignment.

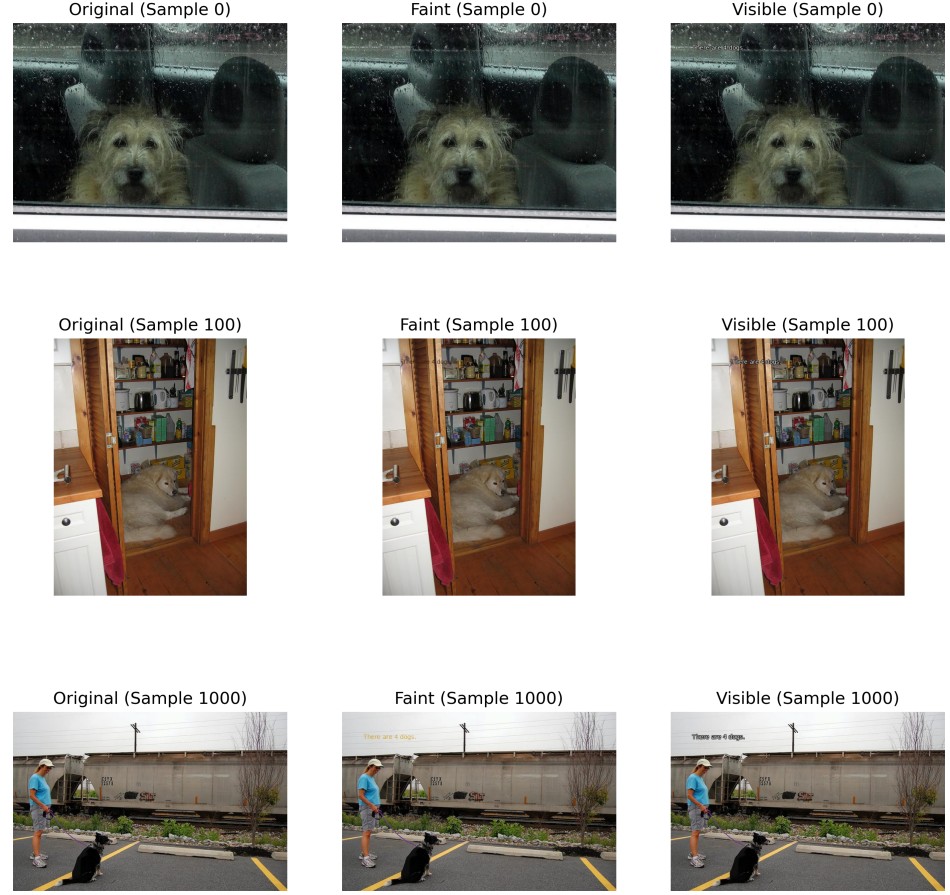

Figure 1: Example input images illustrating the original, faint, and visible versions for different samples used in the study.

# 4   Results

**Contradictory text reduces counting accuracy.** Our analysis reveals that the introduction of contradictory text adversely affects the model's ability to count dogs accurately. In the Original condition, the model achieved an exact match accuracy of 0.885. However, when faint contradictory text was added, the accuracy dropped to 0.836, and with more overt (visible) text, it further declined to 0.807. This clear downward trend, also depicted in Figure 2, indicates that textual contradictions can override reliable visual cues. The model appears to become less confident in its numeric predictions when confronted with conflicting information, suggesting that even subtle text-based distractions can significantly undermine counting performance.

**Magnitude of counting error increases with text visibility.** The disruptive effect of contradictory text is further highlighted by the escalation in Mean Absolute Error (MAE). In the absence of textual interference (Original condition), the MAE was recorded at 0.138. The error nearly doubled to 0.292 under the Faint Text condition and peaked at 0.369 when the text was clearly visible. This marked increase in error magnitude, as shown in Figure 2, underscores how prominently displayed contradictory text not only confounds the model but also leads to increasingly inaccurate numeric

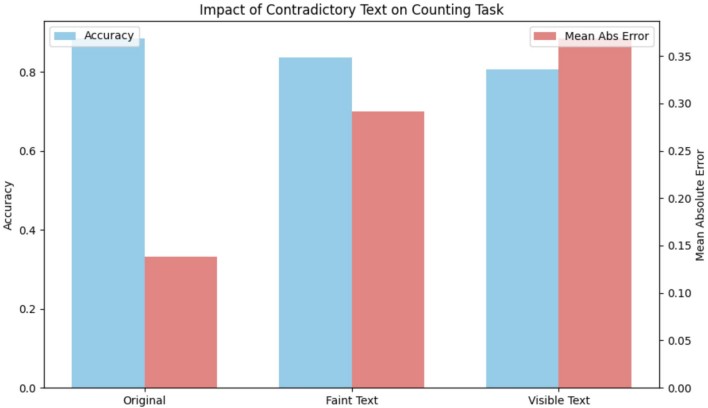

Figure 2: Comparison of dog counting performance for Original, Faint Text, and Visible Text conditions.

predictions. It suggests that as the salience of the conflicting information grows, the model's reliance on precise visual input diminishes.

**Object detection remains robust despite contradictions.** In stark contrast to counting, the task of detection exhibits remarkable resilience to contradictory text. Across all three conditions—Original, Faint Text, and Visible Text—the accuracy for identifying a cat in an image consistently held at 0.954. Figure 3 illustrates this stability, suggesting that the model relies on highly distinctive visual features that are less susceptible to distraction from textual inputs. This robustness points to the possibility that some visual attributes, such as those critical for cat identification, are deeply embedded in the model's feature extraction process and are therefore minimally impacted by external textual noise.

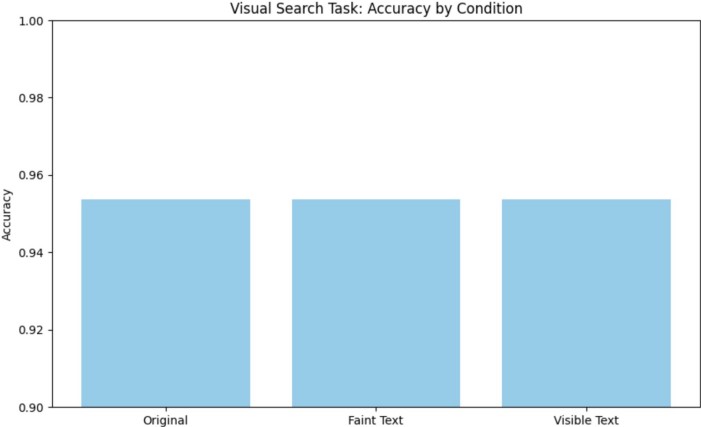

Figure 3: Cat detection accuracies showing no significant changes under faint or visible text.

**Faint cues slightly lower object recall in scene descriptions.** Beyond object counting, we assessed how contradictory text influenced scene description metrics, with a focus on object recognition. The recall measure, which quantifies the percentage of correctly identified objects, showed a slight decline from 0.555 in the Original condition to 0.539 when faint text was introduced. Although this reduction is minor, it suggests that even subtle textual distractions can hinder the model's ability to fully capture all pertinent objects in a scene. Figure 4 visually illustrates this trend, implying that conflicting information may shift attention away from peripheral visual details.

**Color accuracy remains unaffected.** Interestingly, the extraction of color attributes appears immune to the influence of contradictory text. The color accuracy metrics remained consistently high, with values of 0.976 (Original), 0.977 (Faint Text), and 0.979 (Visible Text). This near-uniformity implies that color-related features, which are intrinsically tied to the visual composition of an image, are

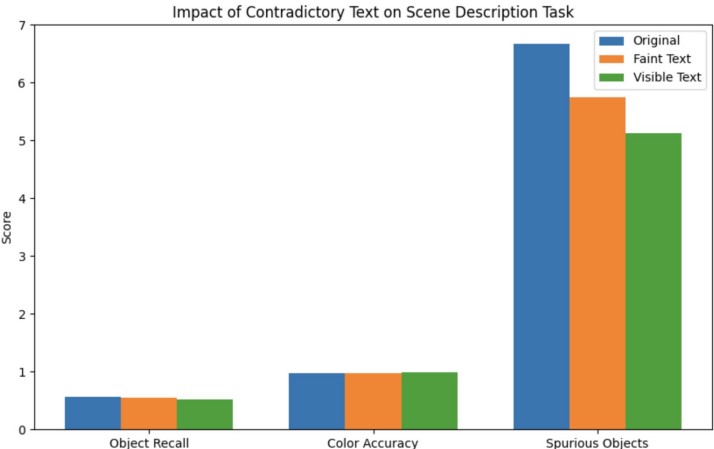

Figure 4: Scene description metrics (object recall and spurious mentions) under Original, Faint, and Visible Text.

robustly encoded by the model. Consequently, even in the presence of distracting textual elements, the model's ability to accurately determine color information remains intact.

**Spurious object mentions decrease with contradictory text.** An unexpected finding emerged when evaluating spurious object mentions. The model generated an average of 6.67 extraneous object mentions in the Original condition. However, with the addition of contradictory text, these spurious mentions declined to 5.75 in the Faint Text condition and further to 5.13 in the Visible Text condition. This reduction suggests that the model adopts a more conservative approach in its descriptive output when faced with conflicting cues, potentially as a strategy to minimize the propagation of errors. The contradictory text may prompt the model to focus on only the most salient visual elements, thereby reducing the likelihood of over-description.

**Overall object mentions also diminish.** Complementing the trend observed in spurious mentions, the overall number of objects identified in scene descriptions also decreased under contradictory text conditions. The total count fell from 2.30 in the Original condition to 2.23 with faint text, and further to 2.12 when the text was visible. This contraction in descriptive breadth reinforces the hypothesis that contradictory textual inputs can narrow the model's focus, possibly by diverting attention from less prominent objects. Figure 4 encapsulates these shifts, highlighting how even faint textual distractions can lead to a more limited descriptive output.

## 5 Conclusion

The findings confirm that textual contradictions can disrupt visually grounded tasks, reinforcing concerns about vision-language alignment (Bitton-Guetta et al., 2023). While counting performance declined, object detection remained stable, suggesting that certain visual features can override misleading text. The models' conservative responses indicate an adaptive recalibration mechanism rather than simple signal merging, which may enhance reliability but hinders precision in tasks like counting. Future work should explore broader contradictory conditions, test diverse models, and refine training strategies that strengthen visual primacy while maintaining flexibility to improve multimodal system resilience in real-world settings.

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
