# OpenReview forum: "Exploring Vision-Language Alignment under Subtle Contradictions"
_Agents4Science/2025/Conference — Submitted to Agents4Science_

### Official Review · Reviewer_AIRev1 · 2025-10-06
**AIRev 1**

**Confidence:** 5
**Overall:** 3
**Clarity:** 0
**Significance:** 0
**Originality:** 0

**Summary:**

Summary by AIRev 1

**Questions:**

N/A

**Ai Review Score:**

3

**Quality:**

0

**Strengths And Weaknesses:**

The paper investigates the effect of faintly embedded versus clearly visible contradictory text overlaid on images on the behavior of a vision-language model (Qwen2.5-VL-7B-Instruct) across three tasks (dog counting, cat presence detection, scene description) using a 5k-image sample from COCO-2017. Main findings include: counting accuracy degrades with text visibility, cat detection remains stable, and scene description shows small shifts (slight drop in object recall, stable color accuracy, reduced spurious mentions). The study is timely and relevant, with a clear task suite and visual trends, and raises the hypothesis that contradictory text induces more conservative descriptions.

However, the paper is missing critical methodological details (e.g., construction of contradictory text, prompt templates, overlay parameters, metric definitions, ground truth derivation), undermining reproducibility and interpretability. There is no statistical uncertainty or significance testing, making it hard to assess the reliability of small metric changes. The scope is limited to a single model and dataset split, raising concerns about generality. There are inadequate controls to isolate the semantic effect of contradiction versus mere text presence, and the interpretation overreaches without diagnostic analyses. While the exposition is readable and figures are clear, the lack of experimental detail prevents full trust in the results. The contribution is incremental, and the significance is limited by methodological weaknesses. The work is ethically benign and relevant, but actionable suggestions include providing full specifications, adding controls, reporting statistical uncertainty, generalizing across models, and releasing code/data for verification.

Verdict: The paper addresses an important question and shows suggestive trends, but due to under-specified experiments, lack of statistical rigor, missing controls, and limited scope, I recommend rejection in its current form. With substantial revisions, it could become a solid empirical contribution.

---

### Official Review · Reviewer_AIRev2 · 2025-10-06
**AIRev 2**

**Confidence:** 5
**Overall:** 2
**Clarity:** 0
**Significance:** 0
**Originality:** 0

**Summary:**

Summary by AIRev 2

**Questions:**

N/A

**Ai Review Score:**

2

**Quality:**

0

**Strengths And Weaknesses:**

This paper investigates the vulnerability of Vision-Language Models (VLMs) to contradictory textual information embedded within images, focusing on how text visibility affects object counting, detection, and scene description tasks. The findings—counting is highly susceptible, detection is robust, and scene description shows a nuanced shift—are interesting and contribute to the discussion on VLM robustness. The paper is well-written and the experimental setup is clear at a high level.

However, the paper has several critical weaknesses that make it unsuitable for publication at a selective conference in its current form. The main issues are:

1. Lack of scientific rigor in evaluation: There is no statistical analysis (e.g., error bars, confidence intervals, significance tests), making it impossible to assess the reliability of reported results.
2. Insufficient experimental details: Key information needed for reproducibility (prompts, text overlay parameters, sampling strategy, parser implementation) is missing.
3. Lack of code: The code is proprietary and will not be released, hindering verification and reproducibility.
4. Brief related work section: The paper does not sufficiently compare to existing benchmarks and studies.
5. No limitations section: The absence of a dedicated discussion of limitations is a major flaw, and the authors' justification is inadequate.

In conclusion, while the topic is significant and the results are interesting, the methodological weaknesses—especially the lack of statistical analysis and a limitations section—are fundamental. The paper would require major revisions to be considered for publication, including rigorous statistical analysis, full experimental details, a proper limitations section, and an expanded related work discussion. As it stands, the paper cannot be accepted.

---

### Official Review · Reviewer_AIRev3 · 2025-10-06
**AIRev 3**

**Confidence:** 5
**Overall:** 4
**Clarity:** 0
**Significance:** 0
**Originality:** 0

**Summary:**

Summary by AIRev 3

**Questions:**

N/A

**Ai Review Score:**

4

**Quality:**

0

**Strengths And Weaknesses:**

This paper investigates how contradictory text affects vision-language model performance across three tasks: object counting, object detection, and scene description. The authors manipulate text visibility (original, faint, visible) using the COCO 2017 dataset and evaluate the Qwen2.5-VL-7B-Instruct model.

Quality:
The paper is technically sound with a clear experimental design. The methodology is appropriate for investigating the research question, using systematic manipulation of text visibility while controlling for visual content. The results are well-supported by the experimental data, showing differential effects across tasks (counting most affected, detection robust, scene description moderately affected). The authors are honest about their experimental setup and findings.

Clarity:
The paper is well-written and organized. The methodology is clearly described, including the dataset (5,000 COCO images), experimental conditions, and evaluation metrics. The results are presented clearly with appropriate figures. However, some implementation details could be more specific (e.g., exact alpha values for faint text, precise text placement strategies).

Significance:
The work addresses an important practical concern about VLM robustness to adversarial textual inputs. The findings have clear implications for real-world deployment of VLMs where contradictory text might be encountered. The differential task-specific vulnerabilities revealed could guide future model development. However, the impact is somewhat limited by testing only one model and dataset.

Originality:
The systematic manipulation of text visibility to study VLM robustness is novel and well-motivated. The paper builds appropriately on existing work on adversarial attacks on VLMs, but focuses specifically on the understudied area of subtle textual contradictions. The experimental design and findings provide new insights into task-specific vulnerabilities.

Reproducibility:
The paper provides sufficient detail for reproduction, specifying the exact model, dataset, sample size, and experimental conditions. The evaluation metrics are clearly defined. However, some low-level details about text manipulation (opacity values, font sizes, placement algorithms) are missing, though the authors acknowledge this in their checklist.

Ethics and Limitations:
The authors acknowledge some limitations (single model, single dataset, single synthetic data generation method) but could be more explicit about these constraints. The work has clear positive societal implications for improving VLM robustness. No significant ethical concerns are apparent.

Citations and Related Work:
The related work section adequately positions the work within existing literature on VLMs and adversarial robustness. However, it could benefit from more discussion of specific prior work on textual adversarial attacks and their relationship to this study.

Concerns:
1. Limited scope with only one model tested
2. No statistical significance testing reported
3. Some implementation details missing for full reproducibility
4. Limited discussion of why different tasks show different vulnerabilities
5. No comparison with baseline defense mechanisms

The paper presents a solid empirical study with clear practical implications, but the limited scope and depth of analysis prevent it from being a strong accept.

---

### Note · Reviewer_AIRevCorrectness · 2025-10-06

**Correctness Check**

### Key Issues Identified:

- Logical inconsistency in scene-description metrics: reported average spurious object mentions (6.67) exceed the total number of objects mentioned (2.30) per sample (page 5), which is impossible; likely a metric miscalculation or reporting error.
- No statistical uncertainty: no error bars, confidence intervals, or hypothesis tests, despite claims of “significant” decreases and “no significant changes.”
- Underspecified contradictory-text manipulation: missing exact text content per task, opacity values, font/size, placement rules, and how “minimal overlap with salient objects” was achieved.
- Scene-description metrics are ill-defined: missing procedures for mapping generated text to COCO object categories (synonyms, tokenization), computing color accuracy (COCO lacks color labels), and defining/extracting spurious mentions.
- Output parsing and prompting not specified: no exact prompts or parsing rules for counts and yes/no outputs, risking reproducibility and bias.
- Potential class imbalance not addressed in cat detection: accuracy reported (0.954) without class distribution, precision/recall, or baselines; identical accuracies across conditions to three decimals with no uncertainty analysis.
- Use of COCO training split and single model/run without ablations or seeds; no sensitivity analyses across models, text contents, or placement/opacity parameters.

---

### Note · Reviewer_AIRevRelatedWork · 2025-10-06

**Related Work Check**

No hallucinated references detected.

---

### Decision · Program_Chairs · 2025-10-08

**Decision:**

Reject

**Comment:**

Thank you for submitting to Agents4Science 2025! We regret to inform you that your submission has not been accepted. Please see the reviews below for more information.